# Hypoxia-Inducible Factors and Diabetic Kidney Disease—How Deep Can We Go?

**DOI:** 10.3390/ijms231810413

**Published:** 2022-09-08

**Authors:** Alina Mihaela Stanigut, Camelia Pana, Manuela Enciu, Mariana Deacu, Bogdan Cimpineanu, Liliana Ana Tuta

**Affiliations:** 1Nephrology Department, Faculty of Medicine, “Ovidius” University of Constanta, 1 Universitatii Street, 900470 Constanta, Romania; 2Pathology Department, Faculty of Medicine, “Ovidius” University of Constanta, 1 Universitatii Street, 900470 Constanta, Romania

**Keywords:** inflammation, hypoxia, HIF, fibrosis, diabetic nephropathy

## Abstract

Diabetes is one of the leading causes of chronic kidney disease (CKD), and multiple underlying mechanisms involved in pathogenesis of diabetic nephropathy (DN) have been described. Although various treatments and diagnosis applications are available, DN remains a clinical and economic burden, considering that about 40% of type 2 diabetes patients will develop nephropathy. In the past years, some research found that hypoxia response and hypoxia-inducible factors (HIFs) play critical roles in the pathogenesis of DN. Hypoxia-inducible factors (HIFs) HIF-1, HIF-2, and HIF-3 are the main mediators of metabolic responses to the state of hypoxia, which seems to be the one of the earliest events in the occurrence and progression of diabetic kidney disease (DKD). The abnormal activity of HIFs seems to be of crucial importance in the pathogenesis of diseases, including nephropathies. Studies using transcriptome analysis confirmed by metabolome analysis revealed that HIF stabilizers (HIF-prolyl hydroxylase inhibitors) are novel therapeutic agents used to treat anemia in CKD patients that not only increase endogenous erythropoietin production, but also could act by counteracting the metabolic alterations in incipient diabetic kidney disease and relieve oxidative stress in the renal tissue. In this review, we present the newest data regarding hypoxia response and HIF involvement in the pathogenesis of diabetic nephropathy and new therapeutic insights, starting from improving kidney oxygen homeostasis.

## 1. Introduction

The unanimous specialist’s opinion is currently that diabetic kidney disease (DKD) is the major cause of end-stage kidney disease, as about 40% of diabetic patients will eventually develop DN, a prevalence depending on the population and ethnicity [1,2]. Systemic metabolic disorders such as hyperglycemia and dyslipidemia cause renal metabolism alterations, leading to renal dysfunction including DKD.

Diabetic nephropathy (DN) is characterized by renal hypoxia, increased oxidative stress, and defective nutrient deprivation signaling, which (acting in concert) are poised to cause both activation of HIF-1α and suppression of HIF-2α. This shift in the balance of HIF-1α/HIF-2α activities promotes proinflammatory and pro-fibrotic pathways in glomerular and renal tubular cells.

Hypoxia response is the mechanism for kidneys to adapt to the oxygen deficiency and to survive under different pathological conditions. Normal oxygen supply is essential for various bio-metabolic processes and to maintain biological homeostasis [1,3,4]. Hypoxia occurs due to different factors such as increased oxygen consumption, vascular remodeling, malfunction of microvasculature, impaired oxygen diffusion by extracellular matrix (ECM) accumulation, mitochondrial abnormality, and anemia, which lead to impaired oxygen delivery [1,5,6].

In this review, we discuss compelling evidence for the direct and indirect role of HIF-1 in the pathophysiology of diabetic nephropathy as the main cause of CKD. We present the recent findings of hypoxia research and the implications of HIF-1 in DN with the purpose of stimulating clinical research to discover possible tools for the diagnosis, follow-up, and monitoring of the clinical impacts of new DKD treatments.

In recent years it is recognized that endothelial and capillary disruption plays an early and important role in the development of diabetic kidney disease (DKD) by affecting the balance of oxygen supply/consumption in the tissues. Therefore, hypoxia is considered to be an important factor in the onset and progression of the disease [7]. Studies from recent years give strong arguments for the chronic hypoxia hypothesis, such as the primary pathophysiological pathway in diabetic kidney disease [8,9]. Hypoxia appears due to an imbalance between oxygen delivery and oxygen demand. The oxygen delivery has principal determinant renal perfusion (blood flow per tissue mass), and for the oxygen demand the most important process which leads to oxygen consumption is active sodium reabsorption. In diabetes mellitus there is abnormal oxygen delivery due to hyperglycemia-associated microvascular damage, and an augmented oxygen demand due to increased sodium reabsorption induced by glomerular hyperfiltration (hyperfiltration is secondary to decreased afferent arteriolar resistance induced by vascular factors: hyperinsulinemia, angiotensin [7,10], and nitric oxide, as well as increased efferent arteriolar resistance mediated by endothelin-1, angiotensin-II, or reactive oxygen products) and sodium–glucose cotransporter upregulation [7,11].

Hypoxia-inducible factors (HIFs) HIF-1, HIF-2, and HIF-3 are the main mediators of metabolic hypoxia, and their abnormal activity seems to play an important role in the pathogenesis of diseases, including nephropathies. In hypoxia, HIF degradation is inhibited, which leads to HIF-α accumulation. As a consequence, HIF-α dimerizes with HIF-β and forms functional HIFs that translocate to the nucleus and activate downstream gene transcription. After an episode of acute kidney injury, the kidneys will start an adaptative repair that completely restores the integrity and functions of the damaged tubules. In the case of incomplete repair (“maladaptive repair”), the process will be characterized by persistence of undifferentiated, atrophic tubules and chronic inflammation. Tubulointerstitial hypoxia, by inhibiting tubular remodeling, will induce chronic local inflammation. Tissue hypoxia and HIFs also stimulate other pathogenic processes involved in CKD: aberrant angiogenesis, anemia, and vascular calcification. Finally, activated HIFs integrate multiple signaling networks and stimulate renal fibrosis evolution to end-stage kidney disease [12,13].

As Figure 1 shows, HIFs are heterodimeric helix–loop–helix transcriptional regulatory factors composed of a labile subunit HIF-α and a constitutively expressed beta-subunit (HIF-β).

The cellular level of HIF-α is regulated by oxygen-dependent proteasomal degradation. In acute hypoxic situations, the decrease in oxygen supply blocks specific prolyl hydroxylase domain-containing protein (PHD1/2/3) activity and inhibits the hydroxylation of HIF-α to stabilize it. As a consequence, the stabilized HIF-α can dimerize with HIF-β. This dimer translocates to the nucleus and transactivates target genes. HIFs with their target genes have been proven to function in a variety of processes, such as: fibrosis, cell proliferation, angiogenesis, erythropoiesis and metabolic switch, inflammation, and apoptosis [1,14,15,16,17]. If hypoxia persists, HIF signaling induces an adaptative response in order to reduce oxygen needs and increase oxygen supply with the aim to reach a new balance [18].

There are three subtypes of HIFs due to the three isoforms of alpha-subunits (HIF-1α, HIF-2α, and HIF-3α). HIF-1 and HIF-2 mediate hypoxic gene transactivation.

HIF-1 and HIF-2 are structurally similar [1] but their expression patterns are different, which leads to divergent target gene regulation. HIF-1 is expressed ubiquitously in organs in most cell types, while the expression of HIF-2 is tissue limited and is especially detected in highly vascularized organs and tissues [1,19]. HIF-1 is involved in the initial adaptation process to hypoxia and HIF-2 expression begins after prolonged hypoxia [20].

In the kidney, HIF-1 expression is present in most of the cell types, with the majority of expression in tubular cells from proximal tubules, distal tubules, connecting tubules, and collecting ducts. It plays multiple functions, such as regulating inflammation, fibrosis, apoptosis, and glycolysis in hypoxic kidney diseases [1].

HIF-2 is expressed mainly in peritubular endothelial cells and in interstitial fibroblasts and plays an important role in the regulation and induction of erythropoietin (EPO) production [21]. 

HIF-3 has a distinct structure from HIF-1 and HIF-2 with three splicing isoforms [1]. There are not as much data regarding the function of HIF-3 and about HIF-3 in the kidney. Some studies show that HIF-3 acts as a target gene of HIF-1 and can negatively regulate the activity of HIF-1 and HIF-2 [1,22].

Chronic hypoxia has been demonstrated to be an important signaling pathway driving diabetic kidney disease [1,9]. Emerging evidence indicates that many of the renoprotective benefits of SGLT2 inhibitors may be the result of the action of HIF-1 [1,23,24].

In addition to hypoxia, there are other nonhypoxic factors such as angiotensin (Ang) II, high glucose, transforming growth factor-β (TGF-β), and radical oxygen stress (ROS), all of which activate HIF-1 and induce renal damage in diabetes [1,25,26]. HIF-1 is implicated in the regulation of these mediators, so it has been suggested that there is a feedback loop through which HIF-1 mediates the initiation and progression of renal damage in diabetes [1,26].

## 2. HIFs and Podocyte Dysfunction

HIF-1 is abnormally expressed in the kidneys of DKD patients [27,28]. The HIF-1 system is often activated before tubulointerstitial injury. Chiu et al. proposed the hyperglycemia–HIF pathways. In normoxia or hypoxia conditions hyperglycemia can activate HIFs in different ways: the synthesis of advanced glycation end products, PKC activation, proinflammatory cytokines, mitochondrial ROS, rage signaling, etc. [29]. These result in reduction in HIF-1 degradation, and activation of nuclear factors by damaging the proteasome HIF-1 gene expression under a normal oxygen environment [27,30]. HIF-1α and HIF-2α are upregulated by hypoxia, and HIF-1α plays a pivotal role in ensuring cells adapt to low-oxygen conditions [27]. Elevated HIF-1 during chronic hypoxia contributes to glomerular diseases’ pathology and proteinuria [31]. Podocytes are the glomerulus’ major cell types. Podocyte dysfunction (reflected by reduced expression of podocyte proteins) and a reduced number of podocytes are characteristic for DN in experimental models and humans [32,33,34]. Recently, Conti et al. [35] demonstrated (by scanning electron microscopy (SEM) analysis of the podocyte cytoarchitecture in type 2 diabetic patients with different stages of kidney disease) that in normoalbuminuric subjects, podocytes are normal in structure, including foot processes. In patients with micro-albuminuria there are markers of podocyte injury: podocyte hypertrophy, diffuse foot process effacement, and pseudocysts representing the site of initial cell detachment from the glomerular basement membrane (GBM) [31]. In the late stages of proteinuric DN, podocyte loss and extensively denuded glomerular basement membranes appear [35,36].

Podocyte and endothelial dysfunction play a key role in the pathogenesis and progression of DN [31,37]. Podocyte dysfunction has been considered a major factor in the development of diabetic glomerular disease [32,38].

The latest studies showed that accumulation of HIF-1 in chronic hypoxia induces various lesions: podocyte epithelial–mesenchymal transition (EMT), cytoskeletal derangement, foot process effacement, and slit-diaphragm dysfunction [31].

In a recent study, Huang et al. observed that the expression of inflammatory cytokines such as MCP-1 and TNF-α was increased in the glomeruli from rats treated with Ang II infusion compared with control rats. Increased HIF-1α expression in the glomeruli was also observed in Ang-II-infused rats [39]. In vitro, Ang II upregulated the expression of HIF-1α in podocytes [39]. Furthermore, knockdown of HIF-1α by siRNA decreased the expression of MCP-1 and TNF-α. Moreover, HIF-1α siRNA significantly diminished the Ang-II-induced overexpression of HIF-1α [39]. The results of this study suggest that HIF-1α participates in the inflammatory response process caused by Ang II and that downregulation of HIF-1α may be able to partially protect or reverse inflammatory injury in podocytes [39].

In models of diabetic nephropathy (DN), activation of HIFs resulted in improved function and decreased proteinuria [40]. The study of Matoba et al. showed that, in the db/db mouse model, inhibition of Rho-kinase signaling prevented the upregulation of HIF-1α and its target genes and reduced glomerulosclerosis [41]. 

Epithelial–mesenchymal transition (EMT) of podocytes and their detachment from the underlying glomerular basement membrane represent one of the cellular events in glomerular diseases [31,42,43]. Petermann et al. detected the presence of viable podocytes in urine in an experimental study of diabetic nephropathy, which suggests detachment of intact podocytes [31,44]. Hypoxia is dominant in diabetic tissues and biopsy sections from patients with DN revealed elevated HIF-1α [31,45]. EMT is considered as the main mechanism of podocyte depletion and pathogenesis of DN [42].

Chronic hypoxia can also damage the podocytes by modulating the expression of slit-diaphragm proteins—the slit diaphragm is a barrier against the loss of plasma protein into the glomerular filtrate and regulates the actin cytoskeleton’s dynamics responsible for the shape of podocyte and foot processes [31,46]. Reduced expression of specific slit-diaphragm proteins during hypoxia induced foot process effacement and proteinuria [46].

HIF-1α regulates cytoskeletal rearrangement, and the study of Chang et al. [47] demonstrated that podocytes, in the context of hypoxia, induce HIF-1⍺ and B7-1 expression (B7-1 plays a major role in regulating podocyte stress fibers). Stabilization of HIF-1α in podocytes results in the induction of the targets ZEB2 (zinc finger E-box binding homeobox 2) and TRPC6 (transient receptor potential cation channel, subfamily C, member 6) [48]. Overexpression of TRPC6 leads to calcium accumulation and, secondarily, stress fiber disarrangement and altered shape of podocytes [48]. Overexpression of ZEB2 manifests in loss of slit diaphragm and, as a consequence, foot process effacement. In addition, overexpression of ZEB2 induces decreased expression of E-cadherin, which leads to podocyte detachment/EMT [48].

## 3. HIFs and Renal Fibrosis

One study showed that HIF-1α enhanced epithelial–mesenchymal transition (EMT) in renal epithelial cells in vitro, and genetic ablation of epithelial HIF-1α reduced tubulointerstitial fibrosis in a mouse model of kidney fibrosis [49]. In the same study, increased expression of HIF-1 and its target genes was found in fibrotic areas of micro dissected kidney tissues from DN patients [49].

In another study on hypertensive DN kidneys of mice with renal fibrosis (Jiao et al., 2018) [50], the upregulation of HIF-1α was found. Moreover, treatment with an HIF-1 inhibitor ameliorated mesangial matrix expansion, glomerular hypertrophy, and fibrosis in diabetic OVE26 mice [1,26]. Taking into account in vitro and in vivo experiments, recent studies indicated HIF-1 as a therapeutic target for an SGLT2 inhibitor for DN [23,24,36]. One study showed that an SGLT2 inhibitor reduced hypoxia-induced HIF-1α protein expression and its target genes in cultured tubular epithelial cells [24,36]. The mechanism was the reduction in mitochondrial oxygen consumption. Treatment with the SGLT2 inhibitor of diabetic db/db mice reduced cortical tubular HIF-1α expression, tubular injury, and interstitial fibrosis [24].

Another study showed that in type 2 diabetes, SGLT2 inhibitors enhanced nutrient deprivation signaling. The mechanism is the upregulation of AMPK and sirtuin1 (SIRT1) (a family of seven NADC-dependent proteins with deacetylase activity, of which SIRT1’s role seems to be the most important for the regulation of HIF-1α acetylation level [51,52], which leads to suppression of HIF-1α [23,52]). 

However, it is still controversial whether HIFs are pro-fibrotic or anti-fibrotic.

Several experimental studies [1,49] showed significant discrepancies regarding the role of HIF-1 in renal fibrosis, and the cause is not clear.

The studies showed that HIFs may play different roles in different renal cells. The pharmacological inhibitors affect all cell types in kidneys, while the inhibition of HIFs by genetic methods may only affect specific renal cell types that may be distinctly involved during renal fibrogenesis. In one study, Higgins DF et al. achieved genetic ablation of HIF-1α from renal epithelial cells in mice, which resulted in an attenuation of the progression of tubulointerstitial fibrosis in unilateral ureteral obstruction (UUO) kidneys [49]. In human CKD, an association of renal HIF-1α expression and tubulointerstitial injury was demonstrated [1,49]. Dallatu, M.K. et al. demonstrated (in a rat model of hypertension induced by a high-salt diet and nitric oxide withdrawal) that elevated epithelial HIF-1α levels exacerbate the progression of kidney damage and renal fibrosis [53]. 

All these studies show that activation of HIF-1α signaling in renal epithelial cells may accelerate fibrogenesis in CKD [1].

Several studies have also investigated the roles of HIF-1 and HIF-2 in the abnormalities of glomerular endothelium in CKD. Luo R et al. suggested that increased endothelial HIF-α induced initial glomerular injury, leading to hypertension and progression of renal fibrosis in Ang II-induced hypertensive chronic injured kidneys [54]. Other studies do not sustain a critical role of HIF-1 in glomerular endothelial pathophysiology [1]. An experimental study by Kalucka J et al. revealed that loss of HIF-1α in glomerular endothelial cells increases hypoxic cell death in vitro, but in vivo, HIF-1α expression in endothelial cells in mouse kidneys is detectable but limited. Endothelial-cell-specific ablation of HIF-1α does not have obvious effects on developmental phenotype in the kidney [55].

One study reported that HIF-1 induces gene transcription of collagen prolyl (P4HA1 and P4HA2) and lysyl (PLOD2) hydroxylases in fibroblasts, but it is unclear whether HIFs can transactivate these fibrogenic genes in kidneys. Therefore, HIF-1 may play a role in extracellular matrix remodeling in renal fibrosis by inducing the genes responsible for collagen deposition, collagen fiber alignment, and extracellular matrix stiffening [56].

## 4. HIFs and Epigenetic Regulation

Currently, there is special research interest in epigenetic regulation in renal fibrosis and hypoxia. The involvement of HIFs in epigenetic regulation under CKD condition has not been well studied. Epigenetics means regulation of gene transcription by direct chemical modification of DNA and by modification of proteins that are closely associated with the locus without changing the DNA sequence. Epigenetic regulations include DNA methylation, histone modification, chromosome conformation, microRNA, and long non-coding RNAs. In CKD, DNA methylation is frequently induced and was demonstrated to facilitate the development of renal fibrosis [57]. For example, a study reported histone methylation changed mesangial cells and epithelial cells in diabetic conditions [58]. Another study showed that, in fibrotic kidneys, histone acetylation and histone ubiquitination are also upregulated [59]. Finally, in aging nephropathy, chromatin conformation change is involved in the induction of collagen III, with more recruitment of RNA polymerase II for gene transcription [60]. In addition, multiple microRNAs and long non-coding RNAs promote renal fibrosis development in various CKD conditions. All the presented results indicate the critical role of epigenetic regulation in the pathogenesis of CKD.

Hypoxia has been reported to reduce connective tissue growth factor (CTGF) expression (an extracellular matrix (ECM) stimulator) in human kidney epithelial cells with the involvement of DNA methylation induction [61]. HIFs also act on the regulation of pro-fibrotic microRNA, such as microRNA-155(miR-155) [62], and anti-fibrotic microRNAs, such as miR-29 [1]. There is not yet scientific data that HIFs can modulate histone modification, chromatin change, or long non-coding RNA expression in kidneys. The mechanism for HIFs to regulate renal cell epigenetic change in CKD remains unclear [1].

## 5. HIFs and Vascular Calcifications

Cardiovascular diseases are the principal causes of death in patients with diabetic nephropathy, and vascular calcification is a common complication in DN. There are multiple factors involved in vascular calcification in patients with DN: advanced glycation end products, dyslipidemia, oxidative stress, and disordered mineral metabolism [63].

HIF-1 has been demonstrated to play a major role in vascular smooth muscle cell (VSMC) calcification [64]. In a hypoxic condition during the progression of CKD, VSMC calcification and osteogenic trans-differentiation significantly increase processes. HIF-1 depletion in VSMC can stop the calcification induced by hypoxia and elevate the inorganic phosphate condition. HIF-1 activators can further stimulate inorganic-phosphate-induced calcification. Together, hypoxia can synergize with elevated inorganic phosphate through HIF-1 induction to enhance VSMC osteogenic trans-differentiation [64].

Clinical data also indicate the close relationship between HIFs and vascular calcification. Nuclear factor-κB (NF-κB) plays an important role in osteoblastic differentiation and also in the activation of HIF-2. In their study, Akahory et al. evaluated 50 aortic valve leaflets from patients with aortic stenosis and aortic valve replacement, in comparison with 10 controls with aortic valve leaflets from patients with annuloaortic ectasia (AAE). They used immunohistochemistry to detect NF-κB, vascular endothelial growth factor (VEGF), HIF-2α, and vascular endothelial cells. In the calcified aortic valves, the authors found that the NF-κB and HIF-2 pathway was highly expressed and associated with an increased expression of VEGF [65].

Li G et al. also showed that elevated serum HIF-1α may contribute to coronary artery calcification (CAC) by CT scanning data analysis of 405 asymptomatic patients with type 2 diabetes mellitus, showing 61.1% sensitivity and 87.6% specificity for predicting CAC extent in this special group of patients [66].

All these studies might support the major role of HIF-1α level as an independent risk factor in vascular and valvular calcification, making it a potential therapeutic target for preventing cardiovascular events in CKD patients [1,65,66].

## 6. Melatonin, HIFs, and Diabetic Nephropathy

Melatonin (*N*-acetyl-5-methoxytryptamine), a hormone and an antioxidant, is a derivative of tryptophan occurring in all living organisms.

Melatonin’s nephroprotective properties have been demonstrated, especially in the context of diabetic nephropathy [52,67,68], including previous studies on alloxan diabetic rabbits (model of type 1 diabetes) and ZDF rats (model of type 2 diabetes) [69]. Numerous mechanisms of melatonin action were proposed, but its nephroprotective effects have never been associated with the attenuation of HIF-1 activity. It is rather surprising that HIF-1 is considered one of the most important factors inducing renal fibrosis—the most common symptom of chronic kidney disease (CKD) [70].

Owczarek A. et al. conducted a study of melatonin (MLT) inhibitory action on HIF-1 in renal proximal tubules, fixing together the three processes: (1) melatonin decreases HIF-1α expression; (2) melatonin affects SIRT1 expression; and (3) deacetylation catalyzed by SIRT1 may regulate HIF-1α stability [52].

In kidneys, the precise regulation of HIF-1 activity is of special importance because they are organs of high oxygen demand and very susceptible to hypoxia and hypoxic damage, which is reported to be responsible for the development of numerous renal pathologies [1,13] in acute kidney injury and in CKD, including diabetic kidney disease. HIFs are of special interest due to their role as central mediators of renal tumor risk [71]. Thus, factors modulating the expression of regulatory subunit HIF-1α might be interesting as potential therapeutics. 

In a recent study by Cheng J. et al., it was demonstrated that hypoxia increases the release of ROS and, in the meantime, VEGF expression, due to the positive feedback mechanism that exists between ROS and VEGF with an impact upon viability and angiogenesis of human umbilical vein endothelial cells (HUVECs). The specific anti-angiogenic mechanism of melatonin has not been elucidated yet, but it seems that it could effectively suppress the enhancement of ROS release and VEGF expression, particularly in combination with KC7F2 (a novel small-molecule HIF-1α translation inhibitor). Collectively, the conclusion of this study was that MLT inhibits the viability and angiogenesis of HUVECs via the mechanism of the HIF-1α/VEGF/ROS axis [72].

Melatonin’s renoprotective effect also seems worth studying in the context of DKD [52,73].

## 7. HIF-1 and Mitophagy

Diabetic tubulopathy is one of initial renal injuries which appears in the pathogenesis of DN [74,75]. Renal tubular hypoxia induces pathological changes in the early and advanced stages of DN, causing renal fibrosis. There are many data that suggest that high glucose levels upregulate HIF-1α expression in animal models and human renal proximal tubular cells of type 2 diabetes mellitus nephropathy [74]. However, the precise role of HIF-1α of diabetic nephropathy in the etiopathogenesis remains unclear. 

Mitophagy is a special, selective type of autophagy described for the first time by LeMasters in 2005 [75] and represents a conserved intracellular process that removes damaged organelles, maintains an adequate mitochondrial content, and ensures cellular quality control. Recent studies revealed diminished levels of mitophagy not only in the renal tubular epithelial cells (RTECs), podocytes, and mesangial cells in hyperglycemic mice, but also in renal biopsies from patients with DKD [75]. 

In a recent study by Yu et al. [74], the renal tubular epithelial cells (HK-2) were subjected to high glucose (HG) exposure. The results indicate that HIF-1α may play a role in mitochondrial autophagy in HK-2 cells exposed to the HG environment. Another important result of this study showed that Parkin/PINK1 is a downstream regulatory factor in an HIF-1α-related mitophagy. The results of this study demonstrate also that mitophagy mediated by HIF-1α-Parkin/PINK1 protects renal tubular cells from apoptosis and ROS production when exposed to HG conditions. 

The study also showed that IL-1b and IL-18 (well-known members of the proinflammatory cytokine family) were significantly increased in the high glucose group, and inhibition of HIF-1α could substantially promote the release of these inflammatory factors. The authors observed that NAC reversed the proinflammatory effect associated with the HIF-1α inhibitor [74]. The results of the study pointed out that the expression of FN (fibronectin) and α-SMA (alpha-smooth muscle actin) were significantly upregulated, and E-cadherin expression was decreased under high glucose treatment. This suggests that HK-2 cells underwent epithelial–mesenchymal transition (EMT). The expression of FN and α-SMA was further increased, and E-cadherin expression was decreased after the addition of YC-1 (a specific inhibitor of HIF-1α). This indicates further enhancement of EMT in HK-2 cells. The authors found that NAC-N-Acetyl-L-Cysteine (an ROS scavenger) significantly reversed the pro-EMT effect of the HIF-1α inhibitor in a high glucose environment [74]. 

Therefore, Yu et al. demonstrated that HIF-1α may reduce HG-mediated inflammation in HK-2 cells and suggested that HIF-1α inhibited epithelial–mesenchymal transition (EMT) in HK-2 cells exposed to the HG environment [74]. The study indicated that mitophagy plays an important role in DN. Recent studies have found that there is a correlation between mitophagy and renal tubular disease in DN. The study of Yu et al. found that HIF-1α alleviates HG-mediated renal tubular cell injury by promoting Parkin/PINK1-mediated mitophagy. Currently, little is known on the relationship between HIF-1α and autophagy, especially mitophagy. It is known that HIF-1α plays different roles in many cellular processes (cellular survival, energy homeostasis, autophagic degradation, and angiogenesis). Yu et al. demonstrated that HIF-1α alleviated high-glucose-induced renal tubular cell injury, which led to a decreased cell apoptosis. HIF-1α has a protective effect in regulating mitophagy by promoting the Parkin/PINK1 signaling pathway. HIF-1α mediates a decreased ROS synthesis which may account for the enhanced mitophagic activity. All these results showed a new mechanism of HIF-1α protection against hyperglycemia-induced oxidative injury and enhancement of mitophagic activity, at least partially via Parkin/PINK1 signaling pathway in tubular cells.

The results of this study demonstrate that HG exposure leads to an increased mitophagic activity and the release of large amount of ROS and, also, the authors observed that the NAC administration reduced ROS levels and increased mitophagy under HG conditions, and these effects were augmented by inhibition of HIF-1α. 

Mitophagy clears only damaged or unwanted mitochondria and fuses with lysosomes in order to degrade the redundant mitochondria through different signaling pathways, PINK1/Parkin and Nix/BNIP3 [49], to maintain the balance in reactive oxygen species (ROS) [75,76].

An important mechanism in the pathogenesis of renal tubular injury in DN [77] has been demonstrated to be mitochondrial dysfunction, which has consequences such as: damaged mitochondrial accumulation, excessive ROS production, renal tubular cell injury, and apoptosis [78]. Treatment with mitochondria targeted antioxidants (mitoQ) has been reported to ameliorate mitophagy and tubular injury. Recent studies have shown that HIF-1α-mediated mitophagy has a positive role in acute kidney injury by inhibiting tubular cell apoptosis and ROS production [79]. 

The research regarding the importance of mitophagy represents the basis for future studies on the pathogenesis of DN and provides important information for disease prevention and treatment [74,75,76,77,78,79].

## 8. Assessment of Hypoxia in CKD

In the past years, various methods for evaluation of tissue hypoxia have been proposed.

The use of ***microelectrodes*** is a classical method for hypoxia detection and the gold standard for determining the oxygen tension in living cells and animals [4]. The principle of this technique is based on oxidation–reduction reactions. This method is widely used to determine renal oxygenation, but there are two major disadvantages of this method: it is highly invasive, and microelectrodes measure the partial pressure of oxygen from both microcirculation and bleeding [4]. Thus, this method is not applicable in humans. The disadvantages are overcome by novel modifications which have been developed, such as the use of ***telemetry*** [80]. By this method, renal oxygenation can be monitored for 2 weeks without anesthesia.

Another method is ***urinary oxygen tension measurement*** (by the insertion of electrode-equipped bladder catheters), which reflects renal medullary oxygen tension and is useful for monitoring renal oxygenation in critically ill patients [4,81]. It has the advantage of being easily applied in clinical settings.

Clinicians also employ the use of exogenous markers such as ***pimonidazole, a 2-nitroimidazole probe*** to detect hypoxia. These probes bind to thiols in hypoxic tissues and allow for the visualization of hypoxic regions by immunohistochemistry [4]. There is evidence that pimonidazole requires a more severe hypoxic response for detection compared to the use of endogenous HIF probes [82]. While the use of endogenous markers and exogenous probes is minimally invasive, they are limited by the fact that biopsies are needed to obtain tissue, and they only provide a snapshot of oxygenation at a single time point. In recent years, advancements in noninvasive imaging techniques have improved our ability to accurately detect hypoxia in clinical practice. Most of these noninvasive techniques rely on magnetic resonance imaging (MRI), positron emission tomography (PET), and single photon emission computed tomography (SPECT) imaging of an oxygen-sensitive label [82]. Several ***positron emission tomography (PET) tracers***
***with nitroimidazoles*** have been studied, for example, 18Ffluoroazomycin arabinoside (18F-FAZA) [4,83], which can be administered in clinical settings. A disadvantage of 18F-FAZA is that it is physiologically distributed to even normal kidneys, and it is excreted from the kidney like other PET tracers [4]. In PET imaging, there is limited resolution to distinguish PET probes in the urinary space and intracellular PET probes. PET tracing using nitroimidazoles is potentially preferred because this technique is less invasive and is suitable for use in CKD patients [4].

Hypoxia can be detected by measuring ***endogenous marker HIF-1α and HIF target genes*** (i.e., VEGF and CA9) by immunohistochemistry [4]. Staining for hypoxia biomarkers allows for a more complete image of hypoxia throughout the entire tissue. Classical techniques, such as ***reverse transcription PCR and immunohistochemistry***, require tissue sample preparation.

An HIF is regulated by oxygen-dependent proteolysis by so-called HIF-prolyl hydroxylases. These key enzymes of HIF degradation are oxygen dependent and can be considered cellular oxygen sensors, because their activity varies in the range of physiologic/pathologic oxygen tensions. Jason R. Tuckerman et al. [84] derived an assay that could measure ***the specific activity of HIF-prolyl*** ***hydroxylases*** and could be applied to small quantities of unpurified enzymes in biological samples. This research showed, using a synthetic hydroxylated HIF-1α peptide for calibration and methods for enzyme quantification, that the von Hippel–Lindau tumor suppressor protein (pVHL, which in normoxia binds HIFs and induces degradation) capture assay provides a method for measurement of specific activity with sufficient sensitivity to measure PHD activity in crude cell extracts. This assay can also be utilized to measure the specific activity of endogenous HIF-prolyl hydroxylase in mammalian cell extracts, to compare the specific activities of different PHD isoforms and their response to hypoxia, and to compare the activity of different recombinant enzyme preparations [84].

The research of Lamia Heikal et al. [85] optimized an assay for HIF-1α to be applied to in vitro and in vivo applications, and used this assay to assess the release kinetics of HIF-1α after endothelial injury. In vitro, a standardized injury was induced in a monolayer of rat aortic endothelial cells (RAECs) and intracellular HIF-1α was measured at intervals over 24 h. In vivo, a rat angioplasty model was used. The right carotid artery was injured using a 2F Fogarty balloon catheter. HIF-1α was measured in the plasma and in the arterial tissue (0, 1, 2, 3, and 5 days after injury). ***An ELISA for the measurement of HIF-1******α***
***in cell culture medium and plasma*** was optimized, and the assay was used to determine the best conditions for sample collection and storage. The results of the ELISA were validated using Western blotting and immunohistochemistry (IHC) [85].

Common detection modalities of oxygenation and changes in O2 consumption rate (OCR) in cell sensing are phosphorescence intensity and ratiometric and lifetime measurements. The study of Alina V. Kondrashina [86] presented a new cell-penetrating phosphorescent nanosensor material called the MM2 probe. This probe provides efficient delivery into the cell and subsequent sensing and high-resolution imaging of cellular O2 in different detection modalities, such as ratiometric intensity and ***phosphorescence***lifetime-based sensing under one-photon and two-photon excitations. The analytical performance of MM2 is demonstrated in physiological experiments with adherent cells and neurospheres representing 2D and 3D respiring objects and detection on time-resolved fluorescent readers, confocal and multiphoton microscopes, and customized ***microsecond fluorescence/phosphorescence lifetime imaging microscopy (FLIM) systems*** [86].

In the past years novel renal ***magnetic resonance imaging*** (***MRI)*** tools have been utilized to evaluate renal perfusion, oxygenation, and fibrosis noninvasively, offering very important data for the pathogenesis of DKD. Assessing renal oxygenation is highly relevant to clinical practice. Renal magnetic resonance techniques provide noninvasive information on renal volume, perfusion, oxygenation, function, metabolism, and microstructural alterations without the need for exogenous contrast media [7].

***Arterial spin labeling (ASL) MRI*** applies a radiofrequency pulse to the circulating aortic blood on its course toward the renal arteries. This process generates a magnetization inversion effect resulting in magnetically labeled arterial blood water that can be utilized as an endogenous tracer. This technique detects small changes in the renal microvasculature [7]. A recent study [87] enrolled diabetic patients with CKD (mean estimated GFR of 51 mL/min/1.73 m^2^) and analyzed them for cortical perfusion, diffusion, and oxygenation. The results revealed a significant decrease in renal cortical perfusion in diabetic patients and moderately reduced GFR, compared to healthy subjects. The study showed an association of cortical perfusion and medullary response to furosemide with annual loss of kidney function [88]. Taking into account the results of this study, ASL can detect incipient and subclinical lesions and can predict DKD progression [7,87].

***Blood-oxygen-level-dependent (BOLD) contrast* *MRI*** is a noninvasive method to evaluate renal tissue oxygenation on the basis of the paramagnetic properties of deoxyhemoglobin. Vinovskis, C. et al. demonstrated (utilizing *BOLD-MRI*) in a recent study on type 1 diabetic patients an association of relative renal hypoxia with renal plasma flow, increased albuminuria, fat mass, and insulin resistance [88]. Another recent study in CKD subjects (42% were diabetic patients) revealed that BOLD-MRI can predict long-term CKD progression [89]. In early DKD, BOLD-MRI detected hypoxic medullary changes, which sustained an increased oxygen consumption in the medulla due to oxidative stress during diabetes [90].

The ***dynamic nuclear polarization MRI technique*** was recently used to study the pathophysiological role of renal hypoxia in diabetes [91].

Recently, a multicenter clinical trial (iBEAt study—NCT03716401) which followed-up 500 patients with type 2 diabetes for 3 years and studied different categories and features of DKD was initiated. The trial can discover prognostic ***imaging biomarkers for DKD***. The study is part of the BEAt-DKD project and aims to examine the gradual changes in imaging biomarkers as kidney failure progresses in the same subject. One objective is to validate imaging *biomarkers* in comparison with histological features and against ***^15^O water positron emission tomography*** [92]. Further studies are still needed to validate these MRI techniques.

## 9. HIFs in the New Therapeutic Era of DKD

In recent years, various therapeutic agents targeting fibrosis have been investigated for diabetic nephropathy treatment, and some clinical trials have been conducted.

Drugs that target renal tubules, podocytes, or vasculature, such as SGLT2 inhibitors [9,23] and DPP-4 inhibitors, as well as drugs that can modulate HIF activity, also inhibiting apoptosis occurring in hypoxic conditions, may lead to next-generation therapeutic drugs that could efficiently control diabetes complications in the kidney [36].

Recent studies showed renoprotective benefits of SGLT2 inhibitors not only by their hemodynamic effects (inhibition of sodium reabsorption in the proximal renal tubule, via tubuloglomerular feedback, reduces the afferent arteriolar vasodilatation induced by hyperglycemia and decreases glomerular hyperfiltration), but also by acting on nonhemodynamic pathogenic mechanisms. This hypothesis occurred after the observations that these drugs are reducing renal events even in patients with advanced DKD. Because of the parallel effect of SGLT2 inhibitors in increasing erythropoietin production and reducing the decline in glomerular function, the question raised by scientists was regarding the possible action of SGLT2 inhibitors on HIFs, which might contribute to their consistent renoprotective effects [23]. 

The oral hypoxia-inducible factor (HIF) prolyl hydroxylase inhibitors are potential alternatives to the ESA (erythropoietin-stimulating agents) approach for the treatment of anemia in patients with CKD. These agents stimulate endogenous erythropoietin production by stabilizing the HIF-α subunit, allowing it to dimerize with the HIF-β subunit and to stimulate genes involved in protection against hypoxia, including the erythropoietin gene. HIF-prolyl hydroxylase inhibitors also influence iron homeostasis through effects on transferrin, transferrin receptor expression, hepcidin, and other iron-related proteins [93] (Figure 2).

This drug class takes advantage of a normal regulatory pathway that boosts hemoglobin production in response to hypoxia by increasing endogenous erythropoietin, improving iron availability, and reducing hepcidin. HIF-PHI agents mobilize stored iron, giving them the “remarkable” ability to be as effective for raising hemoglobin in patients who are not iron replete as in those who are. Another distinction from the ESAs is that HIF-PHI agents can be effective in patients with higher levels of inflammation marked by higher C-reactive protein (CRP) levels. One advantage of PHD inhibitors (or HIF stabilizers) compared to ESA treatment is that these agents are administered orally, which is very helpful for both non-dialysis and peritoneal dialysis patients [1,94]. Furthermore, HIF stabilizers induce physiological EPO production, not like ESA therapies that may cause abnormal and harmful high-peak EPO.

There are also some disadvantages or limitations of HIF-PHI administration. First, HIFs and hypoxia are involved in the regulation of different biological processes. Persistent HIF activation during long periods of time may cause extra side effects: changes in glucose levels, fat, and cholesterol metabolism and the possible promotion of tumor growth. Moreover, because HIFs have different pathological roles in distinct renal cells in CKD, it is difficult to prevent the potential adverse effects on the development of renal fibrosis, angiogenesis, and vascular calcification by global activation of HIFs. Due to these facts, despite the promising outcome of targeting HIF therapy in certain CKD and related complications, there is necessary further research in order to delineate the possible adverse effects and design more specific therapeutic strategies [1].

Five agents in the HIF-PHI class are already on the market in Japan or China, including roxadustat. Studies using transcriptome analysis, confirmed by metabolome analysis, revealed that enarodustat, one of novel therapeutic agents against anemia in chronic kidney disease from the class of HIF stabilizers (HIF-prolyl hydroxylase inhibitors), that acts by increasing endogenous erythropoietin production, could serve to counteract the alterations in incipient diabetic renal metabolism and relieve oxidative stress in renal tissue [94].

Roxadustat, one of the HIF-prolyl hydroxylase inhibitors, designed for renal anemia, has been shown to inhibit HIFs and alleviate the progression of renal fibrosis. Li et al. designed a randomized, multicenter, and active-controlled study (NCT02652806) of the efficacy of roxadustat in treating CKD-associated anemia and implemented animal research to find out if the same drug may retard the progression of fibrosis by regulating Akt/GSK-3b-dependent Nrf2 activation [95].

Robert Provenzano et al. [96] evaluated roxadustat efficacy in non-dialysis-dependent (NDD) evolving to dialysis-dependent (DD) CKD from six pivotal Phase III studies which included 4277 non-dialysis-dependent and 3890 dialysis-dependent patients, with a subset of 1530 who began dialysis 2 weeks to 4 months before randomization to roxadustat or epoetin alfa (defined as recent-onset dialysis). The conclusion was that the efficacy of roxadustat vs. placebo/epoetin alfa for improving Hb level and reducing rescue therapy/IV iron use was demonstrated across the NDD and DD stages of CKD progression.

Singh A. et al. evaluated in two studies (ASCEND-ND (Anemia Studies in Chronic Kidney Disease: Erythropoiesis via a Novel Prolyl Hydroxylase Inhibitor Daprodustat–Non-Dialysis) [97] and ASCEND–D (for dialysis) [98]) a new member of the class daprodustat regarding the efficacy and safety of the HIF-prolyl hydroxylase inhibitor daprodustat. Therefore, as compared with the conventional ESA darbepoetin alfa, in patients with CKD who were not undergoing dialysis and patients with CKD undergoing dialysis, daprodustat was noninferior to ESAs regarding the change in the hemoglobin level from baseline and cardiovascular outcomes in both type of patients.

Despite all these results, because of their potential for increasing angiogenesis and risk of accelerated tumor growth, elevated cancer rates could be a serious concern during treatment with HIF-PHI. Currently, unresolved concerns remain about the long-term safety of roxadustat and other HIF-PHIs as well, for other potential adverse effects including risk for triggering cancers (i.e., neuroendocrine tumors), pulmonary hypertension, thromboembolic events, or progression of CKD. This is the reason the FDA has not yet approved the use of roxadustat or daprodustat in the US. In both cases, the agency cited unresolved concerns about safety as its rationale for denying the applications. Roxadustat is approved in EU countries, including Romania, and more studies in groups of patients with DKD are going to be published in the years to come.

In spite of all these studies, there is still a lack of clinical evidence to support the anti-fibrotic effect of HIF-PHI [23].

As we mentioned before, there are five HIF-PHIs that have completed Phase III trials and are now being marketed and prescribed in clinical practice worldwide: daprodustat, enarodustat, molidustat, roxadustat, and vadadustat, both for dialysis or predialysis patients [99], while new drugs are under pre-clinical study, such as desidustat (ZYAN1) and JNJ-429045343. Above the recognized class effects, each of these five molecules have a different structure, half-life, diversity in PH selectivity, and variable adverse events. In Table 1, we summarize the characteristics of each drug class and the mechanisms and relevant clinical studies of HIF-PHI that were included in the study groups of diabetic NDD-CKD patients [99,100,101,102,103,104,105,106,107].

Abbreviation: AC, active controlled; CERA, continuous erythropoietin receptor activator (epoetin beta pegol); CKD, chronic kidney disease; DA, darbepoetin alfa; DAPRO, daprodustat; DB, double blind; DM, diabetes mellitus; DN, diabetic nephropathy; ENARO, enarodustat;, end of treatment; ESA, erythropoiesis-stimulating agent; Hb, hemoglobin; MOLI, molidustat; NC, noncomparative; NDD, non-dialysis dependent; NR, not reported; OL, open label; PBO, placebo; PC, PBO controlled; QD, once daily; R, randomized; ROXA, roxadustat; TIW, 3 times weekly; VADA, vadadustat; ESA-naïve is defined as no use of ESA for a study-defined time period before start of study.

In the end, we must underline one of the important concerns in diabetic patients, that stabilization of HIFs may exacerbate diabetic retinopathy, a common complication of DM, especially in dialysis patients. HIF-PH inhibitors may increase the expression of VEGF and angiogenesis through the activation of HIF, and it is well-known that HIF-1 and VEGF are closely related to the development and progression of diabetic retinopathy, retinal vein thrombosis, and age-related macular degeneration [100]. 

## 10. Conclusions

Diabetic nephropathy is a major complication of diabetes and a leading cause of end-stage renal disease. It is a complex multifactorial disease, which involves several pathophysiological pathways leading to inflammation and, finally, to fibrosis. Hypoxia-inducible factor HIF-1 is a pivotal molecule that plays an important role in a hypoxic environment. Diabetic kidney disease is characterized by increased activation of HIF-1α, combined with HIF-2α suppression that may importantly contribute to glomerular and renal tubular dysfunction, in apparition and progression of renal disease. Further studies are needed for different classes of drugs, starting from melatonin to HIF-prolyl hydroxylase inhibitors and SGLT2 inhibitors, to show their impact on DKD occurrence and progression by improving kidney oxygen homeostasis.

## Figures and Tables

**Figure 1 ijms-23-10413-f001:**
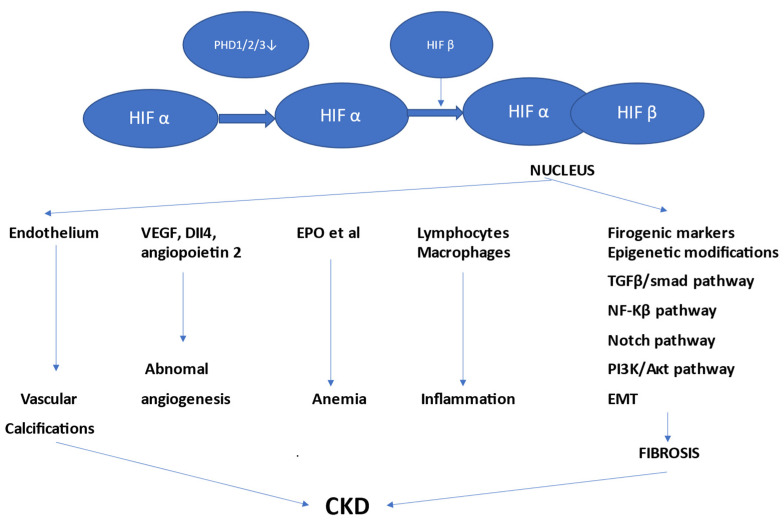
HIF regulation in CKD (adapted from Liu [1]). EMT: epithelial–mesenchymal transition, EPO: erythropoietin, PHD1/2/3: prolyl hydroxylase domain-containing protein, VEGF: vascular endothelial growth factor, TGF: transforming growth factor.

**Figure 2 ijms-23-10413-f002:**
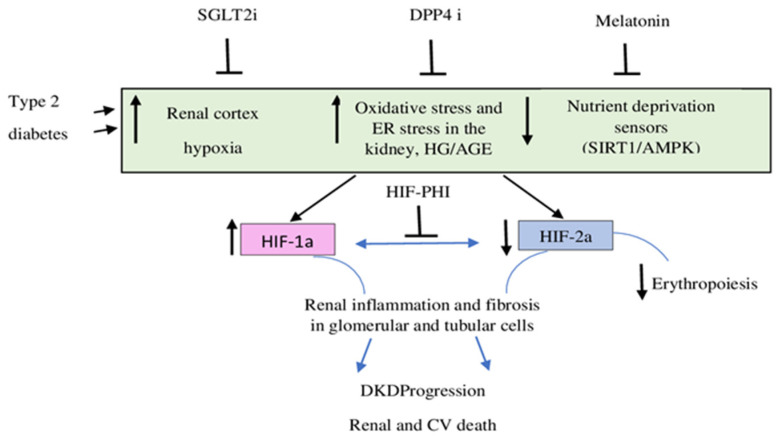
HIFs as triggers of diabetic nephropathy and potential effects of SGLT2i, DPP4i, melatonin, and HIF-PHI in reducing DKD progression (adapted from Packer M. [23]).

**Table 1 ijms-23-10413-t001:** Summary of the characteristics of each drug class and the mechanisms and relevant clinical studies of HIF-PHI in diabetic NDD-CKD patients (adapted with courtesy from V. Haase [99]).

Type of HIF-PH	Mechanisms [100]	Study/Main Author	Study Design: No. of Patients/ DM/DN Patients	Treatment, Duration	Primary Outcomes: Hb Response Rate
**Daprodustat** **(GSK-1278863**	- inhibits PHD2 and PHD3 - absorption 65% - urinary excretion rate < 0.05% - half-life: 1–7 h - metabolized by CYP2C8, UGT1A9	(NCT02791763) Kimura et al., 2019 [103]	R, OL, AC; ESA-naïve and ESA-treated; *n* = 299 **Diabetes mellitus** DAPRO: *n* = 64 (44%) CERA: n = 69 (46%)	DAPRO QD vs. CERA, 52 weeks	Hb at target (11–13 g/dL) during weeks 40–52: DAPRO: 92% CERA: 92%
		ASCEND-ND (NCT02876835) Singh AK et al., 2021 [97]	R, OL, AC; ESA-naïve and ESA-treated; *n* = 3872 **Diabetes mellitus** DAPRO: *n* = 1076 (55.5%) DA: *n* = 1118 (57.8%)	DAPRO QD vs. DA, 52 weeks	Hb at target (11–13 g/dL) during weeks 28–52: DAPRO: 92% CERA: 92%
**Enarodustat** **(JTZ-951)**	- inhibits HIF-PH 1–3, but no effect on various receptors and enzymes - absorption 41.7% - urinary excretion rate: 27–61% - half-life: 11 h - less susceptible to metabolization	(Japic CTI-183870) Akizawa et al., 2021 [101]	R, OL, AC; ESA status NR; *n* = 216 **Diabetic nephropathy:** DAPRO: *n* = 30 (30.9%) DA: *n* = 32 (33.3%)	ENARO QD vs. DA, 24 weeks	Hb at target (10–12 g/dL) during weeks 1–24: ENARO: 89.6% DA: 90.6%
**Molidustat** **(BAY 85-3934)**	- mainly inhibits PHD2 - absorption 59% - urinary excretion rate: 4% - half-life: 4–10 h - metabolized by UGT1A1/1A9	MIYABI ND-C (NCT03350321) Yamamoto et al., 2019 [102]	R, OL, AC; ESA-naïve; *n* = 162 **Diabetic nephropathy** MOLI: *n* = 34 (41.5%) DA: *n* = 22 (27.5%)	MOLI 5–200 mg QD vs. DA; 52 weeks	Hb at target (11–13 g/dL) during weeks 30–36: MOLI: 68.3% (exception weeks 34 and 40 in low eGFR group < 15 mL/min) DA: 80.5%
**Roxadustat** **(FG -4592)**	- inhibits all three PHDs - absorption 40–80% - urinary excretion rate: 1% - half-life 12–15 h - metabolized by CYP2C8/UGT1A9	NCT02652819 Chen et al., 2019 [93]	R, OL, AC; ESA-treated; *n* = 152 (101 ROXA/51 EPO) **Diabetes mellitus** ROXA: *n* = 22 (22%) EPO: *n* = 16 (31%)	ROXA 100 or 120 mg TIW vs. EPO alfa, 26 weeks	During weeks 23–27: ROXA: 92.5% EPO alfa: 92.5%
		NCT02964936A Akizawa et al., 2020 [103]	R, OL, NC; ESA-naïve; *n* = 99 **Diabetes mellitus** *n* = 28 (28.3%)	ROXA 50 or 70 mg TIW, 24 weeks	Hb at target ≥10 g/dL: ROXA (50 mg): 97.0% ROXA (70 mg): 100.0% Hb at target ≥ 10.5 g/dL: ROXA 50 mg: 94.9% ROXA 70 mg: 98.0%
		NCT02988973A Akizawa et al., 2022 [103] Post hoc analysis	R, OL, AC; ESA-treated; *n* = 201 **Diabetes mellitus** *n* = 105 (52.2%)	ROXA 52 weeks	Maintenance of Hb at target (10–12 g/dL)
		ALPS (NCT01887600) Shutov et al., 2021 [104]	R, DB, PC; ESA-naïve; *n* = 594 **Diabetic nephropathy** ROXA: *n* = 109 (27.9%) PCB: *n* = 66 (32.5%)	ROXA vs. PBO, 52–104 weeks	During weeks 1–24: ROXA: 79.2% PBO: 9.9% (*p* = 0.0001)
		ANDES (NCT01750190) Coyne D.W. et al., 2021 [105]	R, DB, PC; ESA-naïve; *n* = 922 **Diabetes mellitus** ROXA: *n*= 398 (64.6%) PBO: *n*= 200 (65.4%)	ROXA TIWc vs. PBO, 52 weeks	During weeks 1–24: ROXA: 86.0% PBO: 6.6% (*p* = 0.0007)
		DOLOMITES (NCT02021318) Barrat et al., 2021 [106]	R, OL, AC; ESA-naïve; *n* = 616 **Diabetic nephropathy** ROXA: *n* = 109 (33.7%) DA: *n* = 98 (33.4%)	ROXA TIW vs. DA, 104 weeks	During weeks 1–24: ROXA: 89.5% DA: 78.0%
**Vadadustat** **(AKB-6548)**	- inhibits all PHD isozymes PHD1-3 - absorption > 75% - urinary excretion rate < 1% - half-life 4–7 h - metabolized by UGT1A1/1A9	PRO2TECT Study group (NCT02648347) Chertow et al., 2021 [107]	R, OL, AC; ESA-naïve; *n* = 1751 **Diabetes mellitus** VADA: *n* = 581 (66.1%) DA: *n* = 599 (68.7%)	VADA QD vs. DA, 52 weeks	Hb at target (10–11 g/dL) during weeks 24–36: VADA: 56.1% DA: 54.9% during weeks 40–52: VADA: 55.3% DA: 54.9%
		PRO2TECT Study group (NCT02680574) Chertow et al., 2021 [107]	R, OL, AC; ESA-treated; *n* = 1725 **Diabetes mellitus** VADA: *n* = 517 (60%) DA: *n* = 518 (60%)	VADA QD vs. DA, 52 weeks	Hb at target (10–12 g/dL) during weeks 24–36: VADA: 65.2% DA: 64.1% Weeks 40–52: VADA: 64.5% DA: 60.7%

## Data Availability

The study did not report any data.

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
