# Peer review of "Hypoxia-Inducible Factors and Diabetic Kidney Disease—How Deep Can We Go?"

_ijms, 2022, doi:10.3390/ijms231810413_

Round 1
Reviewer 1 Report
The authors wrote an interesting review, describing the role of hypoxia in CKD and discussed the characteristics of oxygenation and metabolism in the kidney. Specifically, the effect of hypoxia on CKD progression was emphasized and CKD-related complications. Finally, the potential therapeutic approaches targeting chronic hypoxia in CKD and its challenges were discussed.
Major,
I suggest the authors to introduce a separate paragraph related to the assessment of hypoxia in CKD. The techniques for evaluation of tissue hypoxia include clinical microelectrode-dependent pimonidazole staining measurements, HIF pathway analyses, and oxygen lifetime two-photon phosphorescence microscopy in live animals and the dependence on the level of oxygenation in the blood magnetic resonance imaging and positron emission tomography computed tomography.
If possible, I recommend including paragraphs related to:
-why different types of cells suffer different response to hypoxia in the kidney?
-are there measurable biomarkers or methods to monitor the early phase response to hypoxia that could predict the outcome of hypoxia-responsive renal injury?
I recommend introducing a table summarizing the characteristics of each drug class discussed, mechanisms, relevant studies.....
Sincerely yours,
Author Response
First of all,all authors would like to thank you for the time spent with the review and your precious advice.
About your first request, we includede a detailed description about the assessment of hypoxia in CKD.
In the end, we added a Table that summarizes the the characteristics of each drug class, mechanisms, relevant clinical studies of HIF- PHI in diabetic NDD-CKD patients.
We attached the revised form of the article.
Thank you!

Reviewer 2 Report
The review of Stanigut et al is aimed to describe the multiple and reciprocal connections standing between hypoxia, hypoxia-inducible factors (HIFs) and diabetic kidney disease. The impact (in a diabetic environment) of HIFs on renal fibrosis, epigenetic regulation, vascular calcifications and mitophagy is clearly described. The potentiality of the new class of HIFs-inhibitors in the prevention and treatment of diabetic nephropathy is clearly defined.
Comments to the manuscript:
1) When we talk about diabetic kidney disease, we immediately refer to glomerular podocytes, the cells responsible for the size-selectivity of the glomerular filtration and, at the same time, the renal target of diabetes. There is no mention, however, in the manuscript of the effect exerted by hypoxia and HIFs on podocytes structure and function. A paragraph should be added to describe and discuss in detail this argument.
2) English should be carefully verified along the entire manuscript.
Author Response
First of all, authors would like to thank you for the time spent with the review and your precious advice.
About your first request, we detailed the effect exerted by hypoxia and HIFs on podocytes structure and function, as you kindly suggested.
Secondly, we tried to check again the English spelling, but we also decided it would be better to access MDPI author services regarding this point.
We attached the revised form of the article.
Thank you very much!
Reviewer 3 Report
The review is comprehensive and relevant. The review article includes a clear and concise abstract. The introduction sets the scene by describing all the recent finding to uncover the perspective of hypoxia response and hypoxia-inducible factors in the pathogenesis of diabetic nephropathy. This key message is conveyed through the text, conclusion and figures.
General comments:
1. Please provide some more information about HIF different roles in different renal cells.
2. The methodology for literature search is not described.
Minor comments:
1. The spell check and grammar should be performed.
Author Response
First of all, authors would like to thank you for the time spent with the review and your precious advice.
About your first request, we detailed the HIF different roles in different renal cells: podocytes, tubulointersitial cells, epithelial, etc/
Secondly, we tried to describe better the aim our study and the methodology for literature research.
Finally, we tried to check again the English spelling, but we also decided it would be better to access MDPI author services regarding this point.
We attached the revised form of the article.
Thank you very much!
Round 2
Reviewer 1 Report
Dear Authors,
I appreciate your interest in the suggestions made, and I hope I've helped improve your article.